# Post-Treatment M2BPGi Level and the Rate of Autotaxin Reduction are Predictive of Hepatocellular Carcinoma Development after Antiviral Therapy in Patients with Chronic Hepatitis C

**DOI:** 10.3390/ijms21124517

**Published:** 2020-06-25

**Authors:** Kazuya Takemura, Etsuko Takizawa, Akihiro Tamori, Mika Nakamae, Hiroshi Kubota, Sawako Uchida-Kobayashi, Masaru Enomoto, Norifumi Kawada, Masayuki Hino

**Affiliations:** 1Department of Central Clinical Laboratory, Osaka City University Hospital, 1-5-7, Asahi-machi, Abeno-ku, Osaka-shi, Osaka 545-8586, Japan; takemura-kazuya-rn@alumni.osaka-u.ac.jp (K.T.); takichan@med.osaka-cu.ac.jp (E.T.); mika-a@med.osaka-cu.ac.jp (M.N.); m1352197@med.osaka-cu.ac.jp (H.K.); hinom@med.osaka-cu.ac.jp (M.H.); 2Department of Hepatology, Graduate School of Medicine, Osaka City University, 1-4-3, Asahi-machi, Abeno-ku, Osaka-shi, Osaka 545-8585, Japan; sawako@med.osaka-cu.ac.jp (S.U.-K.); enomoto-m@med.osaka-cu.ac.jp (M.E.); kawadanori@med.osaka-cu.ac.jp (N.K.); 3Department of Hematology, Graduate School of Medicine, Osaka City University, 1-4-3, Asahi-machi, Abeno-ku, Osaka-shi, Osaka 545-8585, Japan

**Keywords:** autotaxin, direct-acting antivirals, hepatocellular carcinoma, hepatitis C virus, sustained viral response, Wisteria floribunda agglutinin positive Mac-2 binding protein

## Abstract

Patients with chronic hepatitis C virus (HCV) develop hepatocellular carcinoma (HCC) regardless of achieving a sustained viral response (SVR). Because advanced liver fibrosis is a powerful risk factor for HCC, we analyzed the association between autotaxin (ATX), a liver fibrosis marker, and post-SVR HCC development within 3 years after antiviral treatment. We included 670 patients with HCV who received direct-acting antivirals, achieved SVR and were followed up for at least 6 months (270 of them were followed up for 3 years or more). We measured serum ATX levels before treatment and 12/24 weeks after treatment. The diagnosis of HCC was based on imaging modalities, such as dynamic computed tomography and dynamic magnetic resonance imaging and/or liver biopsy. The present study revealed that high levels of serum ATX predicted post-SVR HCC development (area under the receiver operating characteristic: 0.70–0.76). However, Wisteria floribunda agglutinin positive Mac-2 binding protein (M2BPGi), another liver fibrosis marker, was a more useful predictive marker especially post-treatment according to a multivariate analysis. Patients with a high rate of ATX reduction before and after antiviral treatment did not develop HCC regardless of high pretreatment ATX levels. In conclusion, post-treatment M2BPGi level and the combination of pretreatment ATX levels and rate of ATX reduction were useful predictive markers for post-SVR HCC development in patients with chronic HCV infection.

## 1. Introduction

Globally, it is estimated that 71.1 million individuals are chronically infected with hepatitis C virus (HCV), of whom 10–20% will develop liver complications, including decompensated cirrhosis and hepatocellular carcinoma (HCC) [1]. Recently approved direct-acting antivirals (DAAs) can achieve a >95% sustained viral response (SVR) for DAA-naïve patients with HCV [1]. However, it was reported that HCC develops or recurs in patients who achieve SVR [2,3]. In the guidelines, follow-up programs are recommended for SVR patients to detect the development of HCC earlier and to evaluate improvements in liver function [4,5]. Screening is recommended to be conducted every 6 months for high-risk patients and every 3–4 months for extremely high-risk patients [6]. Ultrasonography (US) and serum tumor markers are the primary screening modalities. In addition, dynamic computed tomography (CT) and dynamic magnetic resonance imaging (MRI) were combined with these screening methods. Properly stratifying high-risk patients is important for screening. Previous studies reported that older age, male sex, the lack of a SVR, habitual alcohol intake, higher alpha-fetoprotein levels, higher aspartate aminotransferase (AST) levels, lower platelet counts, and type 2 diabetes mellitus are risk factors [7,8,9]. In a large retrospective cohort study, the highest risk factor for post-SVR HCC development has been reported to be the presence of cirrhosis [10]. Therefore, liver fibrosis markers may be predictive markers for the development of post-SVR HCC. Wisteria floribunda agglutinin positive Mac-2 binding protein (M2BPGi), a glycoprotein marker for liver fibrosis, has been reported to be associated with HCC development [11,12,13].

Autotaxin (ATX) was discovered as an autocrine motility stimulating protein in a conditioned medium from A2058 human melanoma cell cultures [14]. Subsequently, ATX has been reported to have lysophospholipase D activity to generate lysophosphatidic acid (LPA) from lysophospholipids in the blood [15]. What was previously thought to be the function of ATX was actually the function of LPA. LPA is involved in cell migration, cell proliferation, neurogenesis, and angiogenesis [16,17]. In recent studies, serum ATX levels have been reported to correlate with liver fibrosis stage in patients with HCV [18,19], chronic hepatitis B virus (HBV) [20], and non-alcoholic fatty liver disease (NAFLD) [21]. ATX is metabolized by liver sinusoidal endothelial cells, and reduced clearance due to liver fibrosis increases serum ATX levels [22]. In addition, serum ATX levels are higher in women than in men [23].

Interestingly, ATX-LPA signaling has been reported to be associated with HCC development [24,25,26]. Kaffe et al. revealed that hepatocyte-specific ATX deletion in mice attenuated HCC development [24]. In addition, Nakagawa et al. established that histological liver fibrosis was attenuated and HCC development was reduced by an ATX inhibitor (AM063) [25]. Therefore, serum ATX levels could be a more useful predictive marker for post-SVR HCC development.

In the present study, we analyzed the association between serum ATX levels and the development of post-SVR HCC within 3 years after antiviral treatment. In addition, we determined whether ATX or M2BPGi was a more useful predictive marker for post-SVR HCC development.

## 2. Results

### 2.1. Comparisons of ATX and M2BPGi Levels by the Presence or Absence of Post-SVR HCC

We recruited 755 patients with HCV who had received interferon (IFN)-free DAA therapy. The patient selection criteria are shown in Figure 1. First, we analyzed the association between pretreatment ATX and M2BPGi levels and post-SVR HCC development using data from Cohort B. The pretreatment ATX levels in patients with post-SVR HCC were higher than those in patients without post-SVR HCC (median values: males, 1.56 mg/L vs. 1.14 mg/L; females, 2.37 mg/L vs. 1.67 mg/L; Figure 2A,B). The pretreatment M2BPGi levels in patients with post-SVR HCC were higher than those in patients without post-SVR HCC (males, 4.31 cutoff index (C.O.I.) vs. 1.82 C.O.I.; female, 3.67 C.O.I. vs. 2.52 C.O.I.; Figure 3A,B). Next, we examined the relationship between ATX and M2BPGi levels at 12/24 weeks after treatment and post-SVR HCC development. The post-treatment ATX levels were higher in patients with post-SVR HCC than in patients without post-SVR HCC (males, 1.39 mg/L vs. 0.95 mg/L; females, 1.74 mg/L vs. 1.37 mg/L; Figure 2C,D). The post-treatment M2BPGi levels were higher in the post-SVR HCC group (males, 2.43 C.O.I. vs. 1.12 C.O.I.; females, 2.76 C.O.I. vs. 1.35 C.O.I.; Figure 3C,D). 

### 2.2. Predictive Ability for Post-SVR HCC Development within 3 Years after Antiviral Treatment

Next, we analyzed the predictive ability for post-SVR HCC development using the area under the receiver operating characteristics (AUROCs). The AUROCs of pretreatment ATX levels were 0.70 for both male and female patients (Figure 4A,B), and those of pretreatment M2BPGi levels were 0.73 for both male and female patients (Figure 4A,B). The AUROCs of post-treatment ATX levels were 0.76 and 0.74 for male and female patients, respectively (Figure 4C,D), and those of post-treatment M2BPGi levels were 0.82 and 0.78 for male and female patients, respectively (Figure 4C,D). M2BPGi tended to be a more predictive marker than ATX for post-SVR HCC development, but this trend was not statistically significant. We also revealed cutoff values to predict post-SVR HCC development. The cutoff values of pretreatment ATX levels were 1.21 mg/L for male patients (sensitivity, 78.9%; specificity, 65.1%) and 2.26 mg/L for female patients (sensitivity, 57.9%; specificity, 79.9%). The cutoff values of pretreatment M2BPGi levels were 2.28 C.O.I. for male patients (sensitivity, 78.9%; specificity, 61.4%) and 2.23 C.O.I. for female patients (sensitivity, 94.7%; specificity, 45.0%). The cutoff values of post-treatment ATX levels were 1.37 mg/L for male patients (sensitivity, 57.9%; specificity, 94.0%) and 1.73 mg/L for female patients (sensitivity, 57.9%; specificity, 80.5%). The cutoff values of post-treatment M2BPGi levels were 1.89 C.O.I. for male patients (sensitivity, 78.9%; specificity, 78.3%) and 1.35 C.O.I. for female patients (sensitivity, 94.7%; specificity, 50.3%).

### 2.3. Multivariate Analysis of Post-SVR HCC Development

Subsequently, we analyzed whether ATX or M2BPGi levels were useful predictive markers for post-SVR HCC development. We formed two groups using the cutoff values and conducted a multivariate analysis using data from Cohort A (Table 1). The multivariate analysis indicated that post-treatment ATX levels and post-treatment M2BPGi levels were significantly associated with post-SVR HCC development in male patients (hazard ratio, 3.75 and 6.43, respectively; Table 1). In addition, pretreatment M2BPGi levels and post-treatment M2BPGi levels were significantly associated with post-SVR HCC development in female patients (hazard ratio, 11.76 and 13.07, respectively; Table 1).

### 2.4. Cumulative Non-Carcinogenic Rate after Antiviral Treatment

After antiviral treatment, 19 patients (both male and female) developed post-SVR HCC within 3 years. We evaluated the cumulative non-carcinogenic rate for markers that were found to be related in the multivariate analysis. The cumulative non-carcinogenic rates in male patients with post-treatment ATX levels ≥ 1.37 mg/L or post-treatment M2BPGi levels ≥ 1.89 C.O.I. were significantly lower than those in patients with post-treatment ATX levels < 1.37 mg/L or M2BPGi levels < 1.89 C.O.I. (Figure 5A,B). In addition, the cumulative non-carcinogenic rates in female patients with pretreatment M2BPGi levels ≥ 2.23 C.O.I. or post-treatment M2BPGi levels ≥ 1.35 C.O.I. were significantly lower than those in patients with pretreatment M2BPGi levels < 2.23 mg/L or post-treatment WFA+-M2BP levels < 1.35 C.O.I. (Figure 5C,D).

### 2.5. Association between the Rate of ATX Change and Post-SVR HCC Development

Because ATX is a quantitative value and M2BPGi is a semi-quantitative value, we finally focused on the rate of ATX change before and after antiviral treatment. The pretreatment ATX levels and rate of ATX change were plotted, and we drew a regression line using data from Cohort B (Figure 6). The equations of the regression lines were as follows: y = −17.61x + 7.38 for males (R^2^ = 0.43) and y = −14.61x + 9.68 for females (R^2^ = 0.41). We divided the patients into two groups using the regression lines. The patients above the regression line were designated as Group A (group with a small reduction in ATX levels before and after antiviral treatment), and those below the regression line were Group B (group with a significant reduction in ATX levels). The proportion of both male and female patients with post-SVR HCC were significantly greater in group A than in group B (Table 2), and the trend was more pronounced in patients without a history of HCC (no HCC vs. first-onset; Table 2). Among patients with a history of HCC, post-SVR HCC also developed in Group B (no relapse vs. relapse; Table 2).

## 3. Discussion

In the present study, we revealed an association between serum ATX levels and the development of post-SVR HCC within 3 years after antiviral treatment. First, we revealed that serum ATX levels before treatment and 12/24 weeks after antiviral treatment in patients with post-SVR HCC were higher than those without post-SVR HCC (Figure 2). The median values of pretreatment ATX levels in patients with post-SVR HCC were 1.56 mg/L in male patients and 2.37 mg/L in female patients, and these values were close to the cutoff values for predicting liver cirrhosis from reagent’s attached document (male, 1.69 mg/L; female, 2.12 mg/L). Advanced hepatic cirrhosis is a main risk factor of HCC [10], and our results seemed to support this theory.

Next, we performed ROC analyses to predict post-SVR HCC development (Figure 4). AUROCs of serum ATX levels before treatment and 12/24 weeks after treatment were 0.7–0.76, and it was thought that ATX levels could stratify a high-risk group of post-SVR HCC patients to some extent. In addition, we revealed that AUROCs of M2BPGi were 0.73–0.82. Comparisons of the ability to predict post-SVR HCC showed that M2BPGi tended to be a better predictor than ATX, but this trend was not statistically significant. However, post-treatment ATX and post-treatment M2BPGi levels had higher predictive performance than pretreatment levels. It was suggested that the presence of HCV infection and/or hepatitis may affect the predictive capacity of post-SVR HCC development.

Previous studies reported that M2BPGi was associated with post-SVR HCC development [11,12,13]. However, studies reporting an association between ATX levels and post-SVR HCC development are rare. Therefore, we analyzed whether ATX or M2BPGi was a higher predictive capacity for post-SVR HCC development, and we revealed that M2BPGi was a more useful marker than ATX (Table 1). M2BPGi has also been reported to be a useful marker for HBV-related HCC [27,28], and NAFLD-related HCC [29]. It is unclear how M2BPGi is involved in HCC development, but it is suggested that it does more than simply indicate liver fibrosis.

M2BPGi is a semi-quantitative value [30]. We witnessed an approximate 10% change in C.O.I. when changing the reagent or calibrator lot (data not shown). Therefore, it is inappropriate to discuss differences of detailed value or degrees of change. In fact, when we analyzed the association between the rate of M2BPGi change and post-SVR HCC development, we found a significant difference only in female patients (data not shown). Because the levels of ATX is a quantitative value, we determined the rate of ATX change before and after antiviral treatment. There was a weak negative correlation between pretreatment ATX levels and rate of ATX change (Figure 6). Despite patients with high pretreatment ATX levels having a high risk of HCC development, all six male patients with ATX reduction rates of 43% or greater did not develop HCC more than 3 years after antiviral treatment (Figure 6A). In addition, 14 out of 15 female patients with ATX reduction rates of 35% or greater did not develop HCC (Figure 6B). According to a previous study, IFN-free DAA therapy decreases serum ATX levels [31]. The present study revealed that ATX changes varied from case to case and that the risk of post-SVR HCC development could be assessed by focusing on ATX changes (Figure 6 and Table 2). Several studies have reported on factors affecting serum ATX levels. The reference value of ATX in women is higher than that in men [23], and even higher in pregnant women [32]. Patients with follicular lymphoma have high ATX levels [33]. However, the patients with kidney disease, heart disease, and diabetes have few changes in ATX levels [34]. The determinations of factors that influence serum ATX levels will improve our understanding of them as predictive markers of HCC development.

The present study had several limitations. First, we analyzed post-SVR HCC development as early as 3 years after antiviral treatment. Therefore, it is unclear whether ATX can predict longer term carcinogenesis. Second, we did not separately analyze for HCC occurrence or HCC recurrence in the multivariate analysis, although patients with a history of HCC before IFN-free DAA therapy have been reported to have a high rate of HCC recurrence [35], because there were not enough patients with developing HCC to perform a multivariate analysis. Finally, validation analysis was not performed. However, the present study clearly showed the role of ATX in prediction for post-SVR HCC. To solve the problems, a larger number of cohorts with longer observation will be required. In addition, more robust cutoff values and predictive formulas that combining multiple biomarkers could be set by analyzing larger cohorts.

In conclusion, the present study revealed that M2BPGi levels especially 12/24 weeks after antiviral treatment were more useful than ATX levels as a predictive marker for post-SVR development within 3 years. However, the combination of pretreatment ATX levels and the rate of ATX change before and after antiviral treatment has the potential to predict post-SVR carcinogenesis. In the treatment of HCV, it is important to stratify high-risk patients and conduct appropriate HCC surveillance, and these results will help for clarifying prognosis.

## 4. Materials and Methods

### 4.1. Subjects

We recruited 755 patients with HCV who had received IFN-free DAA therapy at Osaka City University Hospital between August 2014 and April 2019. Written informed consent was obtained from all patients prior to DAA treatment. The patient selection criteria are shown in Figure 1. The following patients were excluded and defined as Cohort A (n = 670): patients who did not achieve SVR, those in whom M2BPGi was not measured, those who followed up for less than 6 months, and those who developed HCC within 6 months after antiviral treatment because HCC was thought to be present before antiviral treatment. Patients who followed up for less than 3 years were excluded and defined as Cohort B (n = 270). We diagnosed chronic HCV infection based on the presence of serum HCV antibody and detectable HCV RNA via the real-time PCR method. The clinical characteristics before antiviral treatment is summarized in Table 3. The median follow-up duration after antiviral treatment was 25 months in male and 34 months in female. A total of 41 male patients and 30 female patients received treatment for HCC before HCV treatment. A total of 19 male patients and 19 female patients developed post-SVR HCC within 3 years after antiviral treatment. In comparisons between Cohort A and Cohort B, significant differences in age, M2BPGi, PLT (platelet), and Fibrosis-4 (FIB-4) index in male patients were shown. FIB-4 index was calculated using Sterling’s formula: {age (years) × AST (U/L)} / {platelet count (× 10^9^/L) × √ALT (U/L)}. The differences between the two cohorts might depend on the follow-up period, which the high-risk patients were followed up for longer term. This study was conducted according to the principals of the Declaration of Helsinki and was approved by the Ethics Committee of the Osaka City University Graduate School of Medicine (approval number: 4097).

### 4.2. Measurement of Serum ATX Levels

We measured serum ATX levels before treatment and 12/24 weeks after antiviral treatment. Serum samples were stored at −80℃ until measurement. Before the analysis of ATX levels, serum samples were thawed at room temperature and centrifuged at 3500 g for 5 min. Serum ATX levels were measured by a two-site enzyme immunoassay with an AIA-2000 analyzer (Tosoh Co., Tokyo, Japan).

### 4.3. Laboratory Data

Platelet counts and routine biochemical tests were analyzed with standard procedures. M2BPGi was measured with a chemiluminescent enzyme immunoassay (not quantitative values) [30]. Laboratory data were obtained from the medical records. We used laboratory data within seven days before and after blood samples were collected.

### 4.4. Surveillance after Antiviral Treatment and Diagnosis of HCC

For all patients who achieved SVR, US or dynamic CT with contrast media or gadolinium-ethoxybenzyl-diethylenetriamine pentaacetic acid-enhanced dynamic magnetic resonance imaging (Gd-EOB-DTPA-MRI) was performed every 3 to 6 months. HCC was diagnosed using contrast media and imaging modalities, such as dynamic CT or Gd-EOB-DTPA-MRI, and/or liver biopsy.

### 4.5. Statistical Analysis

All statistical analyses and data visualizations were performed by R software ver. 3.5.3 (The R Foundation for Statistical Computing, Vienna, Austria). Comparisons of serum liver fibrosis markers in patients with and without HCC development were performed with the Wilcoxon rank sum test. Diagnostic capacities were analyzed by the AUROCs. Cutoff values were identified by the Youden index. Factors associated with post-SVR HCC development were analyzed using Cox proportional hazards regression analysis. Kaplan–Meier curves were used to analyze the cumulative non-carcinogenic rate in the post-treatment period, and comparisons of two groups were performed with the log rank test. The regression lines were calculated with the least squares method. The comparisons of two groups separated by regression lines were performed with Fisher’s exact test. A *p*-value of less than 0.05 was considered statistically significant. All the analyses were performed based on the patient’s sex because ATX shows sex-related differences. We used Cox proportional hazards analysis (Table 1) and Kaplan–Meier curves (Figure 5) to analyze cohort A, which included cases whose follow-up was completed within 3 years after antiviral treatment, as the analyses involved a timeframe. The other analyses were performed in cohort B, which included cases followed up for more than 3 years, because of the analyses of carcinogenesis at the point of three years after antiviral treatment.

## Figures and Tables

**Figure 1 ijms-21-04517-f001:**
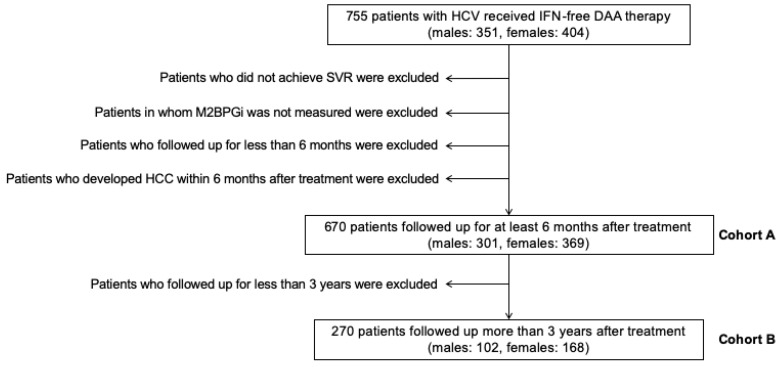
Selection of eligible patients. HCV: hepatitis C virus; IFN: interferon; DAA: direct-acting antiviral; SVR: sustained viral response; M2BPGi: Wisteria floribunda agglutinin positive Mac-2 binding protein; HCC: hepatocellular carcinoma.

**Figure 2 ijms-21-04517-f002:**
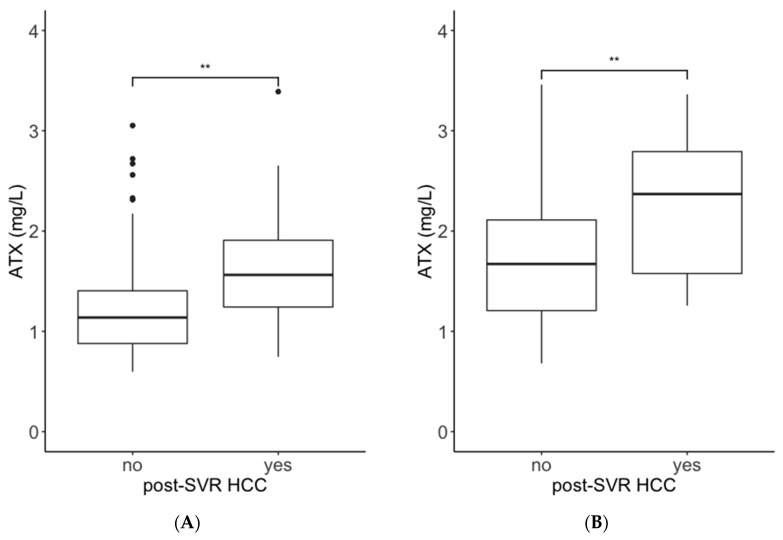
Comparisons of serum autotaxin (ATX) levels in patients with and without post-SVR HCC development. Data from Cohort B were used for the analysis. Data from male patients (**A**,**C**) and female patients (**B**,**D**) are shown. A and B indicate pretreatment levels, and C and D indicate those at 12/24 weeks after antiviral treatment. Boxes represent the interquartile range of the data. The horizontal lines in the boxes indicate the median values. The vertical lines connect the nearest values of 1.5 times the interquartile range from the quartile points. The dots indicate outliers. **: *p* < 0.01; ***: *p* < 0.001. ATX: autotaxin; SVR: sustained viral response; HCC: hepatocellular carcinoma.

**Figure 3 ijms-21-04517-f003:**
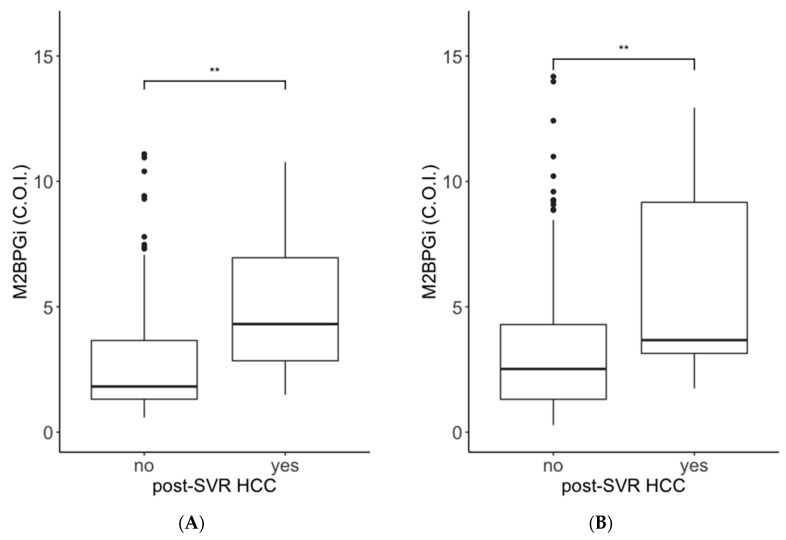
Comparisons of M2BPGi in patients with and without post-SVR HCC development. Data from Cohort B were used for the analysis. Data from male patients (**A**,**C**) and female patients (**B**,**D**) are shown. A and B indicate pretreatment levels, and C and D indicate those at 12/24 weeks after antiviral treatment. Boxes represent the interquartile range of the data. The horizontal lines in the boxes indicate the median values. The vertical lines connect the nearest values of 1.5 times the interquartile range from the quartile points. The dots indicate outliers. **: *p* < 0.01; ***: *p* < 0.001. M2BPGi: Wisteria floribunda agglutinin positive Mac-2 binding protein; SVR: sustained viral response; HCC: hepatocellular carcinoma.

**Figure 4 ijms-21-04517-f004:**
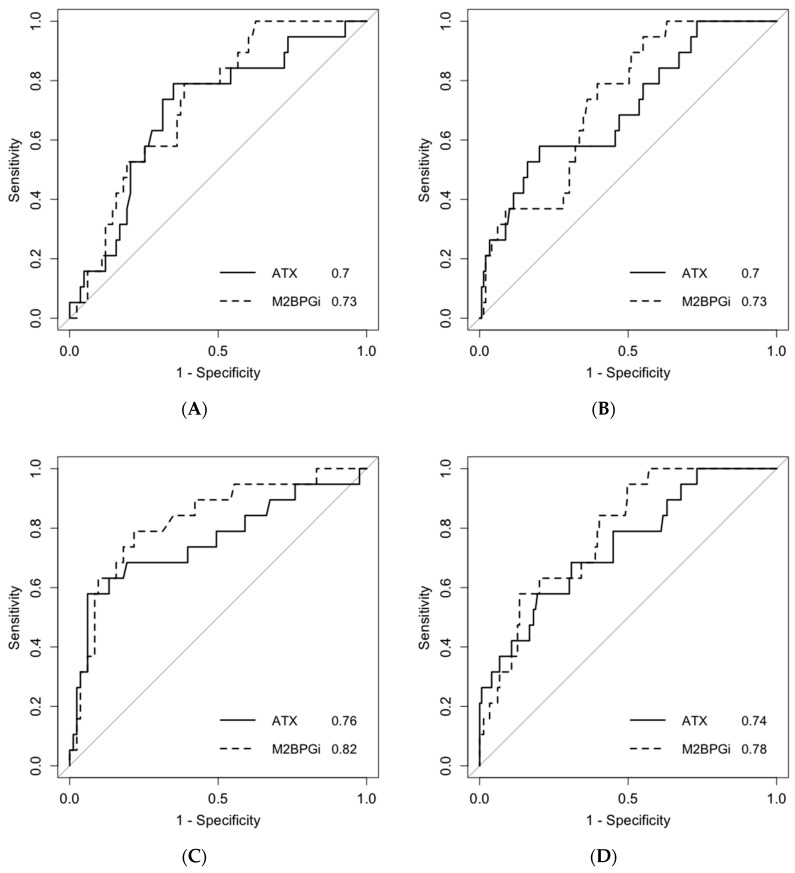
Area under the receiver operating characteristic curves for predicting post-SVR HCC development. Data from Cohort B were used for the analysis. Data from male patients (**A**,**C**) and female patients (**B**,**D**) are shown. A and B indicate pretreatment levels, and C and D indicate those at 12/24 weeks after antiviral treatment. The numbers at the bottom right are the area under the receiver operating characteristic of each liver fibrosis marker. SVR: sustained viral response; HCC: hepatocellular carcinoma; ATX: autotaxin; M2BPGi: Wisteria floribunda agglutinin positive Mac-2 binding protein.

**Figure 5 ijms-21-04517-f005:**
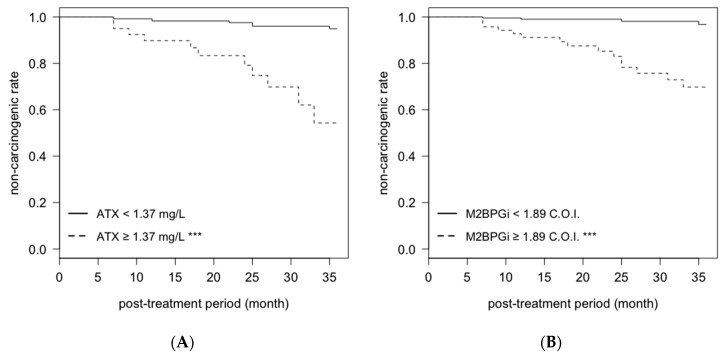
Cumulative non-carcinogenic rate after antiviral treatment. Data from Cohort A were used for the analysis. (**A**,**B**) indicate data by serum levels at 12/24 weeks after antiviral treatment in male patients. Data from female patients before treatment are shown in (**C**), and from female patients at 12/24 weeks after antiviral treatment are shown in (**D**). ***: *p* < 0.001. ATX: autotaxin; M2BPGi: Wisteria floribunda agglutinin positive Mac-2 binding protein.

**Figure 6 ijms-21-04517-f006:**
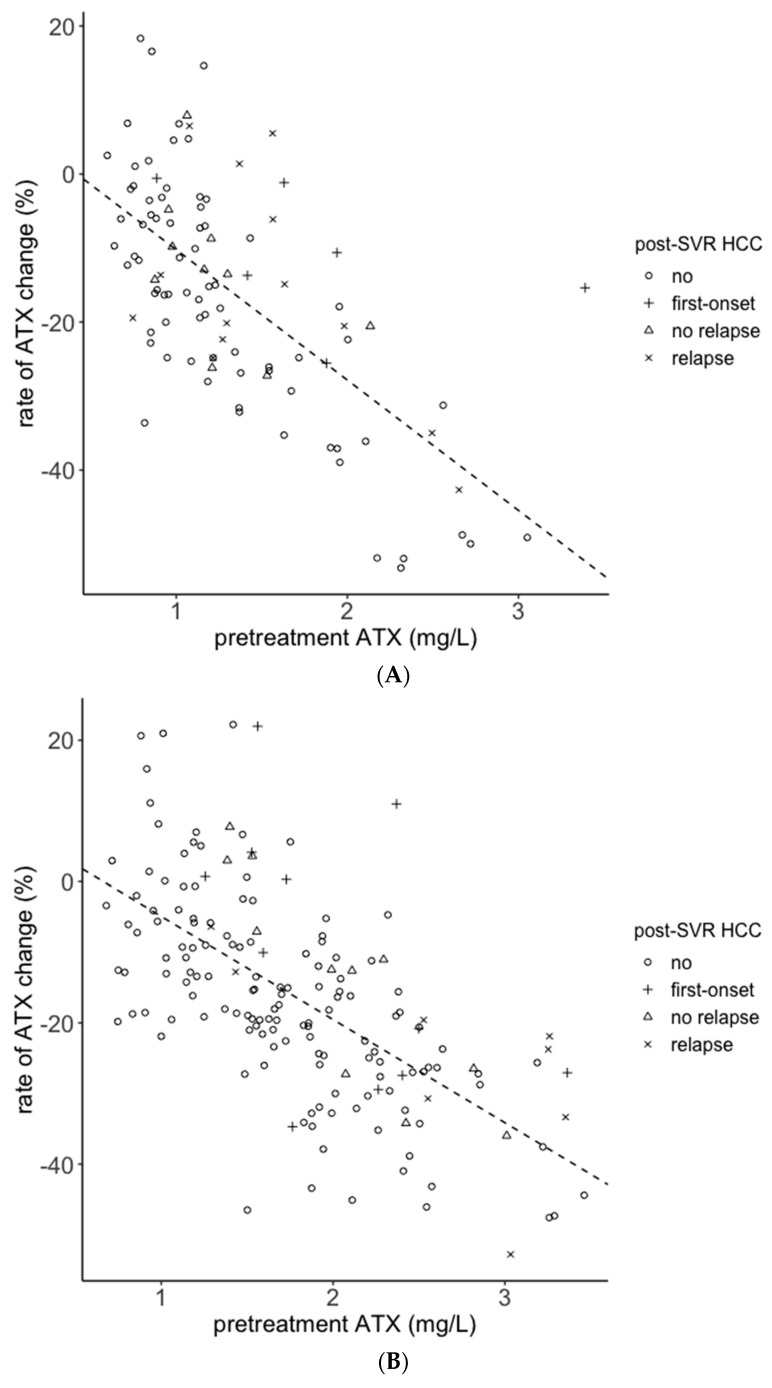
Association between pretreatment ATX levels and the rate of ATX change. Data from Cohort B were used for the analysis. Data from male patients (**A**) and female patients (**B**) are shown. Dotted lines represent the regression lines. no: the patients with no HCC history and no carcinogenesis; first-onset: the patients with no HCC history and post-SVR HCC development; no relapse: the patients with HCC history and no carcinogenesis; relapse: the patients with HCC history and post-SVR HCC development. ATX: autotaxin; SVR: sustained viral response; HCC: hepatocellular carcinoma.

**Table 1 ijms-21-04517-t001:** Factors associated with post-SVR HCC development.

				Multivariate Analysis		
		Factor	Category	Hazard Ratio	95% CI	*p*-Value
male	pretreatment	ATX	≥1.21 mg/L	3.26	0.82–12.89	0.092
		M2BPGi	≥2.28 C.O.I.	2.38	0.60–9.40	0.217
N = 301	post-treatment	ATX	≥1.37 mg/L	3.75	1.32–10.70	0.013
		M2BPGi	≥1.89 C.O.I.	6.43	1.83–22.52	0.004
female	pretreatment	ATX	≥2.26 mg/L	2.50	0.97–6.41	0.057
		M2BPGi	≥2.23 C.O.I.	11.76	1.47–94.07	0.02
N = 369	post-treatment	ATX	≥1.73 mg/L	2.34	0.92–5.97	0.08
		M2BPGi	≥1.35 C.O.I.	13.07	1.66–103.22	0.015

Data from Cohort A were used for the analysis. SVR: sustained viral response; HCC: hepatocellular carcinoma; ATX: autotaxin; M2BPGi: Wisteria floribunda agglutinin positive Mac-2 binding protein; 95% CI: 95% confidence interval.

**Table 2 ijms-21-04517-t002:** Association between the rate of ATX change and post-SVR HCC development.

		No	First-Onset	No Relapse	Relapse	*p*-Value
male N = 102	Group A	28	6	7	7	0.006
	Group B	45	0	3	6	
female N = 168	Group A	57	8	8	5	0.034
	Group B	81	3	3	3	

Data from Cohort B were used for the analysis. ATX: autotaxin; SVR: sustained viral response; HCC: hepatocellular carcinoma; No: the patients with no HCC history and no carcinogenesis; First-onset: the patients with no HCC history and post-SVR HCC development; No relapse: the patients with HCC history and no carcinogenesis; Relapse: the patients with HCC history and post-SVR HCC development; Group A: group with a small reduction in ATX before and after antiviral treatment; Group B: group with a significant reduction in ATX before and after antiviral treatment.

**Table 3 ijms-21-04517-t003:** Baseline characteristics.

	Male			Female		
	Cohort A	Cohort B	*p* Value	Cohort A	Cohort B	*p*-Value
N	301	102	−	369	168	−
age (years)	67 (56–73)	70 (62–75)	0.01	70 (62–76)	71 (62–75)	0.95
therapy régimen ^†^ (A/B/C/D/E/F)	47/102/61/35/6/50	41/39/17/2/2/1	−	68/167/49/31/13/41	58/77/25/3/5/0	−
ATX (mg/L)	1.14 (0.86–1.47)	1.17 (0.91–1.56)	0.28	1.62 (1.20-2.10)	1.71 (1.27–2.25)	0.15
M2BPGi (C.O.I.)	1.82 (1.17–3.62)	2.15 (1.41–4.25)	0.04	2.10 (1.30–4.22)	2.63 (1.46–4.34)	0.10
PLT (×10^4^/μL)	17.1 (12.9–21.8)	14.4 (10.6–20.1)	0.004	16.2 (11.5–21.4)	14.9 (10.7–19.5)	0.06
AST (U/L)	44 (30–68)	47 (33–72)	0.20	40 (28–61)	41 (32–61)	0.27
ALT (U/L)	41 (26–70)	42 (27–70)	0.67	35 (23–54)	37 (27–54)	0.31
FIB-4 index	2.67 (1.77–4.25)	3.28 (2.29-5.33)	<0.001	3.03 (1.86–4.95)	3.51 (2.09–5.26)	0.10
follow-up period (months)	25 (13–39)	−	−	34 (18–42)	−	−
HCC history before HCV treatment (Yes/No)	41/260	23/79	−	30/339	19/149	−
HCC development within 3 years after HCV treatment (Yes/No)	19/282	19/83	−	19/350	19/149	−

Data are presented as the median and interquartile range. Comparisons between Cohort A and Cohort B were performed with the Wilcoxon rank sum test. ATX: autotaxin; M2BPGi: Wisteria floribunda agglutinin positive Ma-2 binding protein; PLT: platelet; AST: aspartate aminotransferase; ALT: alanine aminotransferase; HCC: hepatocellular carcinoma; HCV: hepatitis C virus; ^†^ A: asunaprevir + daclatasvir, B: sofosbuvir + ledipasvir, C: sofosbuvir + ribavirin, D: elbasvir + grazoprevir; E: ombitasvir + paritaprevir + ritonavir, F: pibrentasvir + glecaprevir.

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
