# Peer review of "Post-Treatment M2BPGi Level and the Rate of Autotaxin Reduction are Predictive of Hepatocellular Carcinoma Development after Antiviral Therapy in Patients with Chronic Hepatitis C"

_ijms, 2020, doi:10.3390/ijms21124517_

Round 1
Reviewer 1 Report
The study by Takemura et al. proposed a predictive method to assess risk of HCC development after antiviral treatment of chronic hepatitis C. However, the variation of sensitivity and specificity between gender needs to be validated by more clinical observation. The manuscript is well prepared and data clearly presented to make appropriate conclusion.
Reviewer 2 Report
Takemura et al. submitted a manuscript entitled "Post-treatment M2BPGi Level and the Rate of Autotaxin Reduction are Predictive of Hepatocellular Carcinoma Development after Antiviral Therapy in Patients with Chronic Hepatitis C". The described work calls for questions and comments that are listed below.
The analyses performed alternate between cohorts A and B, separate male and female patients or not, and mixed status and outcomes of HCC (Figure 6).
The abstract should indicate the use of two cohorts (A:670 and B:270 patients) or more accurately two timepoints (6 months and 3 years, with n=670 and n=270).
The authors should clarify their main claim (M2BPGI and/or ATX) and support it clearly with their data and analysis.
How robust are the analyses taking timepoints at 6 months, 1 year, 2 year and 3 years ?
Could a more powerful predictor be built by combining ATX and M2BPOI levels to built ?
Could the authors clarify why M2BPGi is a semi-quantitative value ?
Round 2
Reviewer 2 Report
The revised manuscript is clearer and the authors have included a useful reference.